# Risk Factors Associated with the Prevalence of Upper and Lower Back Pain in Male Underground Coal Miners in Punjab, Pakistan

**DOI:** 10.3390/ijerph17114102

**Published:** 2020-06-09

**Authors:** Madiha Ijaz, Muhammad Akram, Sajid Rashid Ahmad, Kamran Mirza, Falaq Ali Nadeem, Steven M. Thygerson

**Affiliations:** 1College of Earth and Environmental Sciences, University of the Punjab, Lahore 54590, Pakistan; Madihab65@gmail.com (M.I.); drakrampu@gmail.com (M.A.); sajidpu@yahoo.com (S.R.A.); 2Institute of Geology, University of the Punjab, Lahore 54590, Pakistan; Kamran.geo@pu.edu.pk; 3College of Statistical and Actuarial Sciences, University of the Punjab, Lahore 54590, Pakistan; nadeemfalaqali26@gmail.com; 4Department of Public Health, College of Life Sciences, Brigham Young University, Provo, UT 84602, USA

**Keywords:** coal mining, lower and upper back pain, male workers, Nordic musculoskeletal questionnaire, Odd ratio, spinal disorder, ergonomics

## Abstract

There is not enough data available on occupational health and safety issues of underground coal miners in Pakistan. This study focuses on spinal disorders in association with personal and occupational factors. The Nordic Musculoskeletal Questionnaire was used for a cross-sectional study of 260 workers of 20 mines located in four districts of Punjab, Pakistan. Regression models were created for upper back pain and lower back pain of workers whose mean age is 19.8 years (±SD 1.47). Results identify the coal cutting as the most harmful work with odds ratios (ORs) 13.06 (95% confidence interval (CI) 13.7–21.5) for lower back pain and 11.2 (95% CI 3.5–19.4) for upper back pain in participants. Those with greater years of work experience had higher odds of upper back pain (2.4, 95% CI 1.4–3.5) and lower back pain (3.3, 95% CI 1.1–4.4). Number of repetitions (mean value 25.85/minute with ±SD 9.48) are also significant for spinal disorder with ORs of 4.3 (95% CI 3.2–7.4) for lower back and 1.3 (95% CI 1.0–2.4) for upper back. Many other occupational and personal factors are positively associated with the back pain in underground coal mines workers, requiring immediate ergonomic intervention.

## 1. Introduction

Musculoskeletal disorders (MSDs), such as pain, tenderness, paresthesia and any other body discomfort, are reported to be associated with many occupations [1,2]. Pain in the upper and lower back is part of these MSDs [3]. People of every age and gender suffer from disorders in spinal structure and functions [4]. Approximately 15% to 20% of adults suffer from back pain each year [5]. The issue is common in both developing and developed countries of the world [6]. Degenerative spinal disorders are common in many Western countries [7].

The underground mining of coal is a multitask process. Initially, the drilling and blasting is done, to pave the way towards the coal seam [8]. Then, coal is cut by coal cutters, which is dumped and transported to the mine surface [9,10], where it is loaded in vehicles for customers. Workers in these mines frequently report chronic back pain. Some strenuous working conditions require awkward body postures [11]. Exerting muscles repetitively to perform a job, prolonged standing, sitting, or bending upper trunk forward, or folding position of entire body or legs are considered risk factors for spinal disorders [12]. Back pain has become a common reason for job absence, as well as absence from social gatherings. Such conditions affect workers’ efficiency and productivity [13]. Underground coal mining is known for affecting the spinal structure, especially the lower back [14] in workers, because it involves lifting and carrying weight, bending over, kneeling, standing, or sitting for a long time, and adopting awkward body postures for a long time [15]. 

The mines of the study area hire most of their laborers from the hilly areas of Kyber Pakhtunkhwa. The number of workers varies from mine to mine, depending upon the number of active coal faces and also the demand for coal. The hired people mostly inherit this occupation of mining from their ancestors. Training to work in coal mines is very informal—provided by father, brother, uncle, and colleague or through experience—to perform the targeted tasks of underground mining of coal. Workers usually start at age 16. Workers are paid by piece rate and in a bid to earn the maximum wage they work for 13 hours in a day on average with short breaks for tea (called Qehwa) or a meal.

Working seven days a week is very common, since their home villages are distant, and they cannot travel frequently. Only physical illness, fever, cold, cough, injury, or pain forces them to rest for one or two days. The government of Punjab has established hospitals in different areas, particularly for the workers of coal mines, where they get free medical exams and medicines for simple ailments, e.g., temperature, cold, cough, pain, and digestive issues. There is no proper retiring age, and only the inability to do work makes workers retire. The Pakistani government has rules and regulations for the safety and welfare of workers. However, when it comes to the wages, the workers tacitly deal with their contractors to negotiate wages, extra shifts, weekend work, and annual leave.

### Objectives of the Study

What are the tasks (drilling and blasting, coal cutting, dumping, transporting, timbering and supporting, and loading and unloading) of underground coal mining associated with prevalence of low back pain and upper back pain in workers?What are the other factors associated with prevalence of low back pain and upper back pain in coal workers?How can this prevalence be reduced?

## 2. Methodology

### 2.1. Study Area

In the Punjab province, there are four districts, namely Chakwal, Jehlum, Khushab, and Mianwali, which are popular for underground coal mines. Five mines were randomly selected from each district, totaling 20 mines from the four districts. All selected mines are underground and practice traditional methods of coal mining.

### 2.2. Study Population

Prior to initiating the study, researchers conducted walk-through surveys to each mine to observe the various tasks, worker interface during these tasks and hazards associated with these tasks. The number of workers being hired for each task were found to be similar in all mines for each work task, with 45% to 46% of workers at coal cutting, 15% to 16% each at dumping and timbering and supporting, and 7% to 8% of workers at transportation, drilling and blasting and loading separately. 

#### 2.2.1. Cluster Sampling and Inclusion Criterion 

Based on the results of walkthrough survey, the number of participants selected from each task was congruent with the percentage of workers hired by the each of the 20 mines. Using equal allocation ratio technique, a group of 13 workers were selected from each mine (selected randomly from cluster of each task). The sample size is 260 workers. Each group of 13 workers comprised 6 coal cutters, 2 dumpers, 2 workers from timbering and supporting, 1 from drilling and blasting, 1 from transporters and 1 from loaders. Thirteen workers from each mine totaled 260 workers from 20 mines. Only the workers with a minimum of two years’ experience (at the same mine and at the same work task) and with pain/discomfort in the back were requested to participate in the study. 

#### 2.2.2. Verbal Consent of Participation

During walkthrough surveys, the purpose and study methods were explained to workers. Workers were asked verbally (because, according to the local norms, no written statement/agreement was required) for participation in study. Since all the workers with pain in back gave the consent to participate, the response rate was 100%. However, those who did not meet the selection criteria were excluded from the sample prior to the start of study.

#### 2.2.3. Exclusion Criterion

Before the start of survey, an introductory session was made for the workers who gave their consent to participate in the study. In this session, it was openly announced that the workers with no pain in back and less than two years at the same mine doing the same task would be excluded from study participation. There was no count of the workers attending the introductory sessions. Since the objective of the study was to check what factors are responsible for prevalence of pain in back, only the workers meeting the inclusion criterion were assembled in one site, and then the sampling was made from this group. 

### 2.3. Variables 

The 16 factor variables included—the six work tasks (drilling and blasting, coal cutting, dumping, transporting, timbering and supporting and loading and unloading), age, BMI, work experience, overall number of years spent working at underground coal mines, duration of shift, number of repetitions, travel time, part-time work, working days/week and the number of months they work in one year, injury in back triggered by factors not-related to work, and level of pain/discomfort in the body at the end of the day. There were workers who had pain in either upper back, lower back or both. Hence, discomfort or pain in the upper and lower back of the participants were taken as two separate variables dependent on the 16 factors. 

### 2.4. Nordic Musculoskeletal Disorder Questionnaire 

To assess the work-related musculoskeletal disorders (MSDs), the Nordic Musculoskeletal Disorder Questionnaire (NMSDQ) is reported as being a valid and reliable tool [16]. The questionnaire was modified according to the requirements of the underground coal mines, and was translated into the national language of Urdu. The questionnaire was completed by researchers, who were trained to ensure that every participant understood each question, by asking complicated questions in different ways using different terms. The validation of the self-reporting was made by completing the questionnaires on worksites, getting confirmation from the contractors (wherever possible), validating identity by using colleague validation, or seeing their available documents like a National Identity Card, verifying diagnosis reports of prescription from their doctors, and also by observing their physical movements. Additionally, most workers have family that lives or works in the same area. They live and move in groups, so it made the validation of reporting straightforward, because they follow the same routine and are concerned about the well-being of each other. The time to complete the questionnaire varied from respondent to respondent, depending upon their ability to answer. The first part of the questionnaire contained the worker’s name, address, and workplace; the second part focused on socio-economic profile; the third part comprised of questions about work experience (experience of the same task of mining at any site), work routine, facilities at workplace, injury outside workplace (any unintentional or inherited pain in back) etc.; and the fourth part contained questions about pain in the worker’s upper and lower back, in which workers were asked to put the scale on the part of back where they feel pain: those who put the scale on the upper part of back were marked for pain in the upper back, and those who put the scale on the lower part of back were marked for pain in the lower back. The severity of discomfort/pain (little, fairly severe, severe, or very severe) in the body at the end of the day was also asked. Working hours in a day and working months/year were asked to determine the significance of back pain. Body Mass Index (BMI) was calculated by measuring the height and weight of all participants. The frequency of pain in the upper and lower back over a daily, weekly, monthly and yearly basis was asked of the participants, and marked in the relevant sections of the questionnaire.

The number of repetitions performed by every worker of each task were measured by the team of researchers, using stopwatches, based on the use of their respective tools or movement of relevant body parts to perform the task in one minute. For example, the repetitions of a coal cutting worker would be the count of the number of times the worker hammered the coal seam with his cutter in one minute. Likewise, dumping and transporting task repetitions were calculated by counting the number of times they bend and raise their upper trunk (with weight of already mined coal, piled in jute sheet by the workers of coal cutting who spread this sheet after every 10 to 15 kg of cut coal) to fill the haulage, hand trolleys, or sacks hung on the back of donkeys. Repetitions for loading and unloading tasks counted the number of times workers bend their upper trunk, fill the shovel with coal, and throw it to or off of the surface required to be loaded or unloaded, respectively, per minute. Similarly, repetitions for the timbering and supporting tasks counted the number of times workers operate the handsaw and peeler (to prepare timber for support in the mined wings) in one minute, or the number of times they hammer these timbers to fill in the coal seam which was cut by the coal cutters.

### 2.5. Statistical Analysis 

Multivariable logistic regression model was used to analyze data. Overall significance of the model was tested using the Omnibus test and the Hosmer and Lemeshow test. Goodness-of-fit, which identifies how well the research data fits the regression model is essential to check using different tests [17] and is described in Table A1 for model A and Table A3 for model B. Odd Ratios (ORs) and their 95% confidence interval (95% CI) for predictor variables (associated with upper back pain and lower back pain separately) were obtained, to determine the association with upper back pain (in Table A2) and in lower back pain (for Table A4), separately. Two multivariable logistic regression models were made with model A for pain in lower back and model B for pain in upper back. Sixteen personal and occupational factors were fixed as independent variables. In these models, factors not significant at *p* > 0.05 [18] were excluded from the models. Classification tables were also drawn to check the workers’ response rate. The analyses were done using SPSS, IBM, version 22 (Armonk, NY, USA) [19]. See Figure 1 for the description of the process flow for data collection and analysis.

### 2.6. Ethical Approval and Consent to Participate

During this study, the ethical approval was granted by Bioethics Committee, University of the Punjab, Lahore Pakistan (TS-52/CEES). All the participants enthusiastically gave their oral consent to participate in study. According to the local norms, no written document was required to register the consent. 

## 3. Results

### 3.1. Mean Calculation of Physical Traits of Workers and the Percentage Across each Work Task

The mean age of workers was 19.8 years (*n* = 260). The mean travel time to work was 6 min, as most of them lived near or on the worksite (Table 1). The mean for daily working hours 12.63 h. The mean for Body Mass Index (BMI) was 27.43. The mean for number of repetitions (of exertion of muscles to perform the task) was 25.85 per minute with ±SD 90.015, which means that there is wide range of repetitions performed by workers. Some workers perform four repetitions in one minute and some workers perform 40 repetitions per minute. There were four categories of pain (in body at the end of day) which were asked of the participants. The first was “little”, second was “rather severe”, third was “severe”, and fourth was “very severe”. The mean of the reported categories is 2.86, which ~3, which means that the workers have severe pain at the end of day.

### 3.2. Potential Risk Factors for Lower Back Pain 

The Multivariable Logistic Regression (MLR) model (Model A, comprising Table A1 and Table A2) was used to check the significance of 16 independent variables for workers’ lower back pain. Out of these 16, only three factors—travel time, part time work activity, and injury outside the workplace (any accidental or inherited pain in back)—were not significant, which is mentioned separately in Table 2. Only six workers reported injuries not related to their work. The workers who performed second jobs (part time work) after their shift reported that the second jobs only consisted of light work, which showed no positive relation with lower back pain (*p* value = 0.25). 

The Goodness of Fit Model A (Table A1) was checked using the Hosmer and Lemeshow test, with a chi-square value of 2.50 and a *p*-value < 0.001. In Table A2, 13 independent variables were found to be strongly significant for lower back pain. For the task of coal cutting and transporting, the odd ratios (13.06, and 5.21) shows the high level of risk for workers. Therefore, these tasks have a positive and direct association with lower back pain. Risk factors, for pain in lower back, from work tasks dumping, timbering and blasting, and loading and unloading were recorded as 4.49, 2.88, and 1.36, respectively. Out of all six studied work tasks, coal cutting was found very hazardous, with the highest odd ratios (OR) value of 13.06, 95% CI 3.74–21.56, while timbering and supporting (work tasks 6) was observed as the least hazardous, with an OR value of 1.36, 95% CI 1.17–2.73. For the risk factor from worker job experience, number of months worked in a year, hours of working shift and number of repetitions (of exertion or movement of body part to perform work) of task per minute were assessed to be strongly associated with lower back pain (*p* value exactly at 0.00). Also, their values for odds ratios were evaluated to be 2.42, 3.17, 2.44, and 4.38, respectively, which shows their significance for pain in lower back.

### 3.3. Potential Risk Factors for Upper Back Pain

Another MLR model (Model B, comprising Table A3 and Table A4) was used to check different occupational and personal risk factors for upper back pain. A total of 16 independent variables were checked against one dependent variable, i.e., pain in the upper back. Out of 16 variables, five independent variables were of no statistical significance (mentioned separately in Table 3), and 11 were significant for upper back pain (are mentioned in tables of Model B).

Among the variables of no statistical significance, the tasks of dumping (coal from coal seam to the dumping site) and transporting (the coal from dumping site to the outer surface) were not positively associated with pain (*p* value 0.62 and 0.51, respectively). Part-time working activity was found to be of a lighter nature and showed no bodily harm. Contrary to the BMI results of MLR Model A, the results of MLR Model B were not positively associated with pain in the upper back (*p* value 0.113, which was less than 0.05).

Model B from the MLR analysis highlighted the significant factors for workers’ upper back pain. Four out of six tasks of underground coal mining were evaluated to be a cause of the pain. The odds ratios for the task of coal cutting, timbering and supporting, and loading and unloading were 11.24, 4.94, and 5.48 values, respectively, which revealed high levels of risk for upper back pain. Therefore, these work tasks proved positive and direct association with the pain with significance value 0.00. ORs for worker experience, working months in a year, and working hours/day were 2.40, 4.31, and 2.54, respectively, whereas all the variables showed *p* value = 0.00, which highlights their strong significance for the risk of upper back pain. For the number of repetitions per minute, an OR value of 1.35 shows a positive association for pain in upper back. From all of these 11 significant variables, age revealed the least level of risk for upper back, showing an OR value of 1.22.

#### Prevalence of Pain with Respect to Age Groups

Reporting of frequency of pain from workers of different age groups is shown in Figure 2. The participants were asked when the last time they experienced back pain was. The given options to this question were last week, last three months, last six months, or chronically so. The highest number (*n* = 118) of workers is in the second age group (26–35 years) and the largest reporting, from this group, is of the pain during the last week. Out of the 76 workers of first age group (16–25 years), no worker reported chronic pain, 41 reported of last three months of pain, and 21 reported pain in last 6 months. The chronic pain was highest in third age group (*n* = 66), which is from 36 to 45 years of age. 

## 4. Discussion

This cross-sectional study, based on the self-reporting of workers, finds the role of personal and occupational factors in causing pain in upper and lower back of the workers of underground coal mines of Punjab Pakistan, and identified that laborers performing heavy manual duty were more vulnerable to back injury, resulting in chronic back pain [20]. The obtained values from both of the models of regression analysis support that many of the 16 independent variables were responsible for pain in the upper back and the lower back of the workers. The findings of this work are in agreement with previous findings that upper and lower back pain is a common musculoskeletal issue among coal miners, and it is imperative to highlight such risks for developing preventive measures [1,21]. The occupational factors and personal traits of workers in underground coal mines have different potential of risk for such pain. Some factors have direct and positive association, while others are not significant.

The physical traits of workers are positively associated with pain in both the upper and lower back. The current study showed that physical characteristics (weight, height, and resulting BMI data) of workers varied widely, because they belonged to Punjab (where people are of normal or tall height) and to the areas of Khyber Pukhtun Khwa where people are of relatively small height. The mean BMI is high, because there was a wide range of workers’ heights, i.e., 4’9” to 6’2” and weight, from 46 Kg to 98 Kg. Body Mass Index (BMI) has a positive correlation with lower back pain among people of different ages [22]. The BMI of all male workers surveyed in this study were mostly obese, and 35% were overweight, which can be seen in the mean value of BMI in Table 1. Like other studies, this research also finds that BMI plays a role in causing lower back pain. However, this research did not find this factor to be significant (in the results obtained in the MLR Model B, the BMI has a *p*-value > 0.05) for pain in the upper back of the participants. Additionally, it does not have a positive value of the logistic coefficient B, which makes it not significant for the pain in the upper back (Table 3). Biomechanical processes of human body demonstrate that age is somehow a relevant factor for back pain [23,24]. Furthermore, as the workers grow older, they get more physical problems, and they become less efficient in working [25]. The efficiency of mining workers is based on the quality and quantity of work with respect to time. With growing age, workers feel less strength to perform their work task, which is laborious wherever the manual handling of material is done, hence, age plays role in workers’ getting less efficient in work. In this study, most of the workers started working at the age of 14. With years of experience (at an underground mine and at the same task), workers get different levels of work-related musculoskeletal disorders, e.g., frequent, acute, or chronic discomfort in the upper limb, the lower limb, the upper back, or the lower back and, consequently, they become unable to work longer. In this trial, a few workers with ages greater than 40 years were seen on sites of underground coal mining. This is because they start working at a very early age, and do not feel capable of working more hours or more years in the mines. From the obtained data, it was found that young workers are used to working for 13 hours a day, while older workers were unable to work more than 4 or 6 hours a day. The current study shows that upper and lower back pain is closely associated with the advancing age of workers, showing that tissue damage with the passage of age leads to back pain in the older workers. However, it is not easy to differentiate amid pathological degeneration and ordinary variations in coal mine workers because of aging [26].

Several studies report many occupational factors contribute to musculoskeletal disorders among underground coal mine workers [27,28]. The predominant factor leading to back pain is heavy manual lifting, repetitive movements, a faster work pace, inadequate recovery time, static or dynamic body postures of shoulders, elbows or wrists, body vibrations, low temperature, and varying levels of mechanical pressure [29,30,31]. Postural fluctuations and weightlifting are considered the most important concerns among these risk factors [32,33]. In this study, it was established that, due to the working environment, the workers involved in underground coal mining tasks (timbering and supporting, coal cutting, and dumping) were prone to higher risk of back pain than that of surface coal workers (loading and unloading). The workers involved in the tasks of transporting and drilling and blasting were partially underground and partially on the mine surface, so their significance for pain was relevant to the extent of performance.

The different body postures, such as prolonged standing, bending in awkward positions, moving in confined areas, and extensive lifting were commonly among underground coal miners [34]. As in the cited research, similar body postures were also observed in this study. The surface employees’ duties encompass servicing, repairing equipment, and transportation, along with discharging administrative roles at ground tasks. Substantial physical fitness and adequate power is required by underground coal workers to perform their routine duties [35,36]. However, several investigations have been carried out regarding the connection of excessive repetitive movements and muscles-related back injuries [37]. In previous research, strong relationships between excessive repetition and back pain were observed in mining workers [37,38]. This investigation also corroborates these findings among coal mining workers. Moreover, the epidemiological literature reveals that prolonged bending position may trigger injuries in the backs of miners [38]. The workers performing underground mining in a bending position are highly prone to risk of lower and upper back injuries [39]. Improper rest can also prompt back pain in this working population. Similarly, it was found that extreme bending postures in the confined spaces and insufficient rest resulted in lower and upper back pain among coal miners.

The study shows that mining workers felt no discomfort or fatigue prior to the start of the shift, as they lived about 5 min away from workplace. Additionally, no worker with any prior history of back pain was included in the study, so the occupational factors were the reason behind pain in the back of all the workers participated in this study. Thus, the workplace is mostly likely the cause of he back pain experienced by workers. Workers who travel long distances each day to arrive at the mine may experience greater fatigue and resulting back pain, compared to workers who live relatively close to the mine. Thus, this occupation is rightly associated with pain/discomfort in workers who are not exposed to exertion of traveling long distance to their job on daily basis.

Workers get few extended breaks from work each year, which increases the likelihood of increased back stress, which may result in back pain. Workers have to work in confined areas for extended hours, which could inflict back pain in workers [12,40]. Working for almost 13 hours with very short break time makes workers vulnerable to sustaining upper as well as lower back pain. Manual drilling also causes exertion of shoulders and upper back. Manual drilling with forward bend creates awkward back postures, such as twisting. Such awkward movements cause weakness in abdominal and posterior spinal muscles, which can result in chronic back pain. Workers with short break times are more likely to get lower back pain, as compared to workers with longer break times [41]. Working hours, without passive or active breaks, increases the chances of back pain [42], as reported in this study. The results show that underground coal miners rarely take a proper rest to relax their muscles, increasing the risk of back pain. Moreover, working months in a year was also found to be positively associated with pain in the upper and lower back. Additionally, the ORs coded high levels of risk in both MLR models. Insufficient time for complete body muscle recovery was also noted as a problem, in addition to all other physical and psychological burdens [43].

Each work task has a different level of risk for both upper and lower back. Task 2 (coal cutting) has the highest level of risk for pain in upper back. To cut coal from the coal seam, workers have to sit or lie on their back or belly (to adjust themselves in narrow spaces) for long periods, and such awkward postures cause pain in their back [44,45]. Since the task of drilling and blasting has a positive logistic coefficient B and *p*-value < 0.05, it is directly associated with pain in both the upper and lower back, but is less significant for pain in the upper back. The tasks of transporting (the mined coal from the dumping site to the outer surface) and timbering and supporting have lower odds, and the task of dumping is totally not significant for the pain in upper back.

A quick glimpse of both of the MVL models shows that a greater number of the studied factors are risky for lower back pain as compared to upper back pain. All of the six tasks were significant for lower back pain, whereas four were significant for pain in upper back. The task of dumping and transporting were not significant for pain in upper back, because these tasks target only the lower limbs and the lower back. In the dumping and transporting of coal, the exertion of arms and upper limb was less, and workers also had reasonable break times, resulting in less vulnerability of upper back pain. Task 2 (coal cutting) exhibited the highest level of risk for pain in upper back, whereas the tasks of drilling and blasting, transporting, and timbering and supporting were relatively less significant. 

### Limitations of Study 

The study was conducted without any equipment to monitor the awkward movement of the spine or the related illness in back of all study participants. The cross-sectional study methodology also meets the known limitations, including the fact that it cannot analyze behavior over time, and that it does not determine cause and effect. This study included a self-reporting methodology, which has its known limitations. While this study included 260 miners to be representative of all miners in the population, we were limited to 20 mines in each district, making it difficult to determine representative numbers for all known mines in Pakistan.

## 5. Conclusions

The occupational factors of underground coal mines and the personal factors of workers are found to be strongly associated with the pain in the upper and lower back of workers. There was a wide range of number of working hours/day, number of repetitions performed, height, weight, and BMI of workers. Hence, the impact on occurrence and frequency of pain and in the upper and lower back also varied. However, the risk factors for pain in the lower back were greater than the risk factors for pain in the upper back. The frequency of pain was high in the workers of older age, though the majority workers were of a young age. 

Recommendations:

Mining companies should invest in machinery, tools, processes, and training that will reduce the risk factors for back injury;Provide ergonomic training to coal mine workers, especially in proper lifting and maintaining posture;Workers should monitor their health and wellness status, especially in the maintenance of their BMI values;Increase rest time and consider reducing working hours, especially for the aged workers;Establish annual medical evaluations and develop procedures for medical intervention when symptoms are reported.

## Figures and Tables

**Figure 1 ijerph-17-04102-f001:**
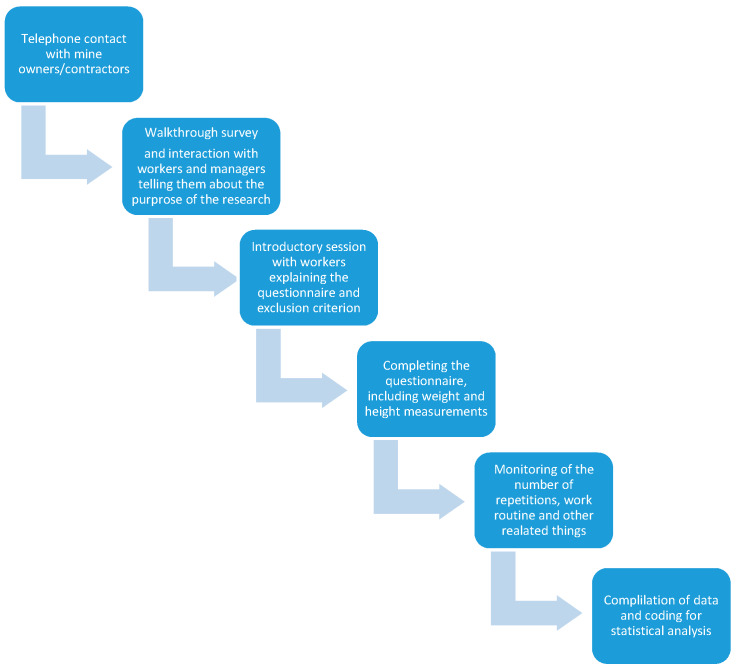
Process Flow Chart.

**Figure 2 ijerph-17-04102-f002:**
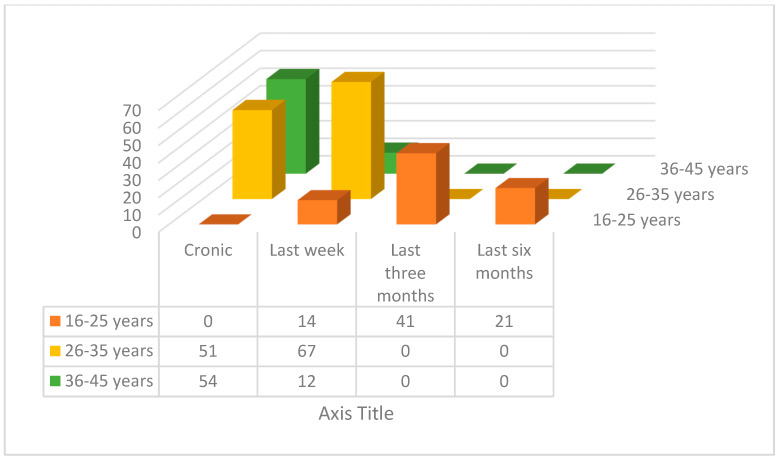
Association between workers’ age and frequency of pain in back.

**Table 1 ijerph-17-04102-t001:** Summary of personal and occupational factors of workers.

Descriptive Statistics	Mean	Standard Deviations
Age (years)	19.80	±1.47
Height	5 ft. 6 in.	±1.130
Weight (kg)	65	±1.7
BMI (kg/m^2^)	27.43	±6.515
Within range i.e., 18.5 to 24.9 (*n* = 45, 17.3 or 17%) <18.5 (*n* = 57, 21.92 or 22 %) ≥25 (*n* = 67, 25.7 or 26%) ≥30 (*n* = 91, 34.61 or 35%)		
Travel time to work place (minutes)	0.06	±0.234
No of working hours/day	12.63	±5.237
No of working months/year	8.43	±2.50
Experience of workers (years)	8.00	±4.086
No. of repetitions /minute	25.85	±9.48
Level of discomfort in body at end of day	2.86 (severe)	±0.8916
Workers with history of nonwork-related injury	0.03	±0.1843

**Table 2 ijerph-17-04102-t002:** The factors not significant for lower back pain.

Factors	Wald	Sig
Travel time	0.322	0.112
Part time working activity	2.54	0.256
Injury outside of workplace	1.36	0.121

**Table 3 ijerph-17-04102-t003:** The factors not significant for upper back pain.

Factors	Wald	Sig
BMI	0.740	0.113
Part time working activity	1.234	0.063
Any Injury outside of the workplace	1.014	0.091
Work stage 3 (Coal Dumping)	8.547	0.621
Work stage 4 (Transporting)	7.245	0.514

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
