# Peer review of "Risk Factors Associated with the Prevalence of Upper and Lower Back Pain in Male Underground Coal Miners in Punjab, Pakistan"

_ijerph, 2020, doi:10.3390/ijerph17114102_

Round 1

Reviewer 1 Report

I thank the authors for addressing the previous comments. My remaining comments are as follows:

Abstract: Line 20-21 "The ORs for their work experience is 2.409 (95% CI 1.389-3.541) for upper back and 3.3 (95% CI 1.1-4.4) for lower back pain." Perhaps this could be "Those with greater years of work experience had higher odds of upper back pain (2.4, 95% CI 1.4-3.5) and lower back pain (3.3, 95% CI 1.1-4.4).

Introduction: Line 38 "Workers of.." should be "Workers in.."

Line 41 what is the difference between bending and folding?

Line 48 should it be "strength" or "numbers"

Line 50 "given" should be "provided"

Line 61 "leaves" should be "leave"

Methodology: Line 83 "was ensured congruent to the" should be "was congruent with the"

Line 83 "For this," could be deleted

This sentence: "For random sampling, no algorithm or simulation process was used because in it, though unknown, chances respondent selection was the same since the total population size (N) was unknown." Is unclear and perhaps could be deleted.

Overall, the paragraph in the "Cluster sampling and inclusion criterion" section is repetitive and could be reorganised to improve readibility

Line 115 "Discomfort or pain in upper and lower back of the participants were the two dependent variables". Does this mean that for the multivariable analysis that upper back pain was compared to lower back pain and lower back pain was compared to upper back pain? Otherwise what is the comparison category? Were there participants who had both?

Line 135 "asked to out scale on the" it is not clear what out scale means - does it mean "point out"

Line 159 has Lameshow, this should be Lemeshow.

Line 163 if the dependent variable is upper back pain compared to lower back pain then this should be stated for clarity ie "the predictor associated with upper back pain compared to lower back pain.

Line 164 has multivariate, this should be multivariable

Table 1 heading "Descriptive of personal and occupational factors of workers". should be "Summary of personal and occupational factors of workers"

What it the category "Within range" in Table 1?

Why is 2.86 identified as severe? What is this based on?

Table 2 has "insignificant" in the heading. This should be not significant, please check the remainder of the document.

There needs to be consistency with decimal points throughout, some places have 1, some 2, some 3.

Line 204 "coefficient B was evaluated as positive.." The use of the coefficient is not of value, and it appears to confuses the interpretation of results throughout. Of interest is the OR.

Line 205 "revealed their significance" this is not correct this is a statistically significant association"

Line 214 "of no significance" should be "not statistically significant"

Line 216-217 "The significance of travel time has p value of 0.041 which is again not positively associated." This is not clear. The use of the term "positively associated" is not clear and it the p value is 0.041 why was it not included in the model?

Line 219 p value 0.0113 for BMI - this is not correct. The p value in table 3 is 0.113 which is more that 0.05.

Line 229 "ORs showed positive values" by definition OR are positive. The wording should be positive association

Discussion: Line 247, "proved that many of the" This is a cross-sectional analysis and causality can't be determined thus there is no proof. There would also need to be comparison with a not pain group doing the same activities to determine if the same characteristics where associated with back pain.

Line 250 - also can't demonstrate that this is common about coal miners as there is no comparison with those who do not have pain.

Line 337 "relatively less significant" this should be "lower odds". The relative nature of work tasks is not examined in this study.

Line 339 "When comparing both MVL models, it became clear that lower back is at more risk of injury as compared to upper back because the risk factors are high in MLR model for upper back" I don't think this statement is correct.

The authors don't really address the stated third objective of "How can this prevalence be reduced?"

Line 528 and line 542 the fitted model equation is incorrect. These are not linear models, they are logistic thus the dependent variable has a binomial distribution. The dependent variable needs to be reported as log(p/1-p) ie the probability of having back pain.

Author Response

On behalf of my co-authors, we appreciate the time taken to conduct this peer-review of our work. These are important comments and we have completed our responses below.

Reviewer #1

I thank the authors for addressing the previous comments. My remaining comments are as follows:

Abstract: Line 20-21 "The ORs for their work experience is 2.409 (95% CI 1.389-3.541) for upper back and 3.3 (95% CI 1.1-4.4) for lower back pain." Perhaps this could be "Those with greater years of work experience had higher odds of upper back pain (2.4, 95% CI 1.4-3.5) and lower back pain (3.3, 95% CI 1.1-4.4). Done

Introduction: Line 38 "Workers of.." should be "Workers in.." done

Line 41 what is the difference between bending and folding? Explained

Line 48 should it be "strength" or "numbers" Replaced

Line 50 "given" should be "provided" Replaced

Line 61 "leaves" should be "leave". Replaced

 Methodology: Line 83 "was ensured congruent to the" should be "was congruent with the" Replaced

Line 83 "For this," could be deleted. Deleted

This sentence: "For random sampling, no algorithm or simulation process was used because in it, though unknown, chances respondent selection was the same since the total population size (N) was unknown." Is unclear and perhaps could be deleted. Deleted

Overall, the paragraph in the "Cluster sampling and inclusion criterion" section is repetitive and could be reorganised to improve readability. Repetition is removed

Line 115 "Discomfort or pain in upper and lower back of the participants were the two dependent variables". Does this mean that for the multivariable analysis that upper back pain was compared to lower back pain and lower back pain was compared to upper back pain? Otherwise what is the comparison category? Were there participants who had both? Explained

Line 135 "asked to out scale on the" it is not clear what out scale means - does it mean "point out". It was ‘’ put the scale’’. Correction is made.

Line 159 has Lameshow, this should be Lemeshow. Corrected

Line 163 if the dependent variable is upper back pain compared to lower back pain then this should be stated for clarity ie "the predictor associated with upper back pain compared to lower back pain. Explanatory sentence added.

Line 164 has multivariate, this should be multivariable. Corrected

Table 1 heading "Descriptive of personal and occupational factors of workers". should be "Summary of personal and occupational factors of workers". Correction is made

What it the category "Within range" in Table 1? Added the correct range

Why is 2.86 identified as severe? What is this based on?  Explanation added in the section There were 4 categories of the pain (in body at the end of day) which were asked from the participants. First was little, second was rather severe, third was sever and 4th was very severe. The mean of the reported categories is 2.86 which ~3 and means that the workers have severe pain at the end of day.

Table 2 has "insignificant" in the heading. This should be not significant, please check the remainder of the document. Checked and replaced

There needs to be consistency with decimal points throughout, some places have 1, some 2, some 3. The 3 decimal points are reduced to two but the values with one cannot be changed. And in tables , these cannot be reduced or increased.

Line 204 "coefficient B was evaluated as positive.." The use of the coefficient is not of value, and it appears to confuses the interpretation of results throughout. Of interest is the OR. It is removed from both of the sections.

Line 205 "revealed their significance" this is not correct this is a statistically significant association" Replaced with word “shows”.

Line 214 "of no significance" should be "not statistically significant" done

Line 216-217 "The significance of travel time has p value of 0.041 which is again not positively associated." This is not clear. The use of the term "positively associated" is not clear and it the p value is 0.041 why was it not included in the model? Correction is made of this error. Thanks for highlighting.

Line 219 p value 0.0113 for BMI - this is not correct. The p value in table 3 is 0.113 which is more that 0.05. Corrected

Line 229 "ORs showed positive values" by definition OR are positive. The wording should be positive association. Corrected

Discussion: Line 247, "proved that many of the" This is a cross-sectional analysis and causality can't be determined thus there is no proof. There would also need to be comparison with a not pain group doing the same activities to determine if the same characteristics where associated with back pain. Explained in the text.

Line 250 - also can't demonstrate that this is common about coal miners as there is no comparison with those who do not have pain. Replaced with proper words wherever the word ‘’demonstrate” is used for the results of our study.

Line 337 "relatively less significant" this should be "lower odds". The relative nature of work tasks is not examined in this study. Replaced

Line 339 "When comparing both MVL models, it became clear that lower back is at more risk of injury as compared to upper back because the risk factors are high in MLR model for upper back" I don't think this statement is correct. Correct explanation is added.

The authors don't really address the stated third objective of "How can this prevalence be reduced?" the recommendations of the study aim at reduction of the problem.

Line 528 and line 542 the fitted model equation is incorrect. These are not linear models, they are logistic thus the dependent variable has a binomial distribution. The dependent variable needs to be reported as log(p/1-p) ie the probability of having back pain.

Since the significance of our regression coefficients was checked at 5% interval of significance, that’s why we wrote “significance at 0.05” in both of the equations in subscripts to elaborate”. Still if required, it can be deleted.

Reviewer 2 Report

thank you for your revisions appropriately done.

Author Response

We appreciate your review of our work and thank you for your time.

This manuscript is a resubmission of an earlier submission. The following is a list of the peer review reports and author responses from that submission.

Round 1

Reviewer 1 Report

This is an association between physically demanding work (like in miners) and back pain, what is obvious and conclusive. Furthermore there is an association between age and back pain. This study seems to prove these associations. Nevertheless i have some comments.

  • Background:
    • this article is due to be published in an international journal. Most readers to not know about the specific working conditions in Pakistan like : when to workers start with working in a mine? Will the Workers be education in skills? Are there any work protection rules or laws? How many hours a day and how many days a week the workers have to work? do they have regularily daily breaks and breaks by holidays? are there rules for sick leave? How the workers are paid? when will the workers retire? This should be described in the introduction.
    • how many workers do work in these mines totally?
  • Methods:
    • the participants were selected randomely. How do you know that the participants were representative and not selected by bias? The mean age was 19,8 years, what is quite young. What is the mean age of all workers in all mines participating?
    • In which tasks workers of the three age clusters work? Did you adjust for task and age?
    • All participants made self reports on their work place and work conditions. Did you perform any validation like on "number of repetitions"? Self reports tend to over- oder underestimate real conditions.
    • Are you sure the participants did understood the questions of the questionnaire?
    • how did you define upper and lower back pain? Did you use sort of body pain drawing for assembling to upper or lower back pain? if so, did the Workers report on further pain sites?
    • did you make a power calculation (why did you select 13 workers from any cluster of each task)?
    • how did you define "nonwork-related back injury"?
    • you excluded painfree workers - why? nevertheless you report on them. How many of all workers addressed were excluded? How many of all workers were pain free?
    • what is "work experience"? how did you qualify?
    • you did not ask for years on work in the mine - why?
    • what is "part-time work"?
    • please give a flow process chart.
  • results
    • you did exclude workers with back injury from outside, but you report on these: "...injury outside workforce were insignificant". Explain or omit.
    • how many workers were aged 16-25 years, 26-35 years, 36-45 years? There were no workers older 45 years? if so, why (please discuss in discussion)? Does back pain increase with aging?
    • did BMI differ by age?
  • discussion
    • please assort better: short summary of main results - discussion of associations and ORs - discussion of specific  conditions in Pakistan 
    • The discussion should pronounce all results which are related to specific working conditions in Pakistan. Please compare to condition in western states and states with work protection rules? Do reports on pain differ?
    • your participants were rather Young. Why? please discuss.
    • mean BMI was overweight. in rather young workers this is surprising. Are young pakistan people overweighted?
    • "The current study showed that physical characteristics of workers varied widely, probably because they belonged to different provinces of Pakistan. " this sentence is not supported by background or results.
    • "were mostly obese and 35% were overweight". This sentence is  not supported by results.
    • "...previous findings that upper and lower back pain is a common musculoskeletal issue among coal miners and it is imperative to highlight such risks for developing preventive measures [1, 21]" the body of literature on this topic is much broader and should be discussed concerning conditions in Pakistan.

    • "...become less efficient in working". This sounds awkword because years on work improves skills and efficiency. Perhaps you want to express something else.
    • "The study shows that mining workers felt no discomfort or fatigue prior to the start of the shift as they lived about 5 minutes away from workplace. Also, no worker with any history of non-work-related back pain were found." This might express false attribution of the workers as common in many studies ("work makes me ill"). Please discuss.
  • Limitations: you only proved associations no causality. It remains unclear whether the participants were representative for all Miners, whether self reports are valid

Reviewer 2 Report

Thank you for the opportunity to review this paper. My comments are as follows:

Abstract: Line 16 I am unsure what "multilateral' means. The sentence could just begin "Regression models.."

Line 18 odd ratio should be odds ratios

Line 22 "of body parts" could probably be deleted to aid clarity.

Introduction: Line 33 "Almost 15-20%..." perhaps should be "Approximately.."

Line 42, "for the absence from job" should be "job absence"

The last couple of sentences prior to the Objectives section are a little repetitive and need reworking.

Line 54 Are you trying to control or reduce prevalence?

Methodology: What is a "walk through survey"? The sentence is also unclear and needs to be reworked.

It appears that clusters were created to reflect the over profile of type of workers? However, this paragraph is not very clear and needs to be reworked.

If clusters were made to reflect the worker profile and it was the same for every mine, why is table 1 required?

Were the questionnaires self-completed or done using an interviewer?

Consent: This sentence "The workers enthusiastically gave consent and no one refused to participate" is subjective and should be deleted.

Line 84 states that those who did not meet selection criteria were excluded. One criteria is stated in line 75 and then exclusion criteria are in line 87-89. This section is a little confusing and needs to be reworked for clarity.

Why were workers who did not have back pain excluded? It may have been useful to have a comparison group. Or they may have had pain in the past that was now resolved? How could there be a comparison done if they are excluded - if doing a logistic regression would the dependent variable be to compare back pain to no back pain?

Line 93, how is "work experience" defined? And how is this different from "working months/years"?

Would workers know the number of repetitions? This should perhaps be linked to the description of how repetitions are determined which is in Line 113 and beyond by saying "number of repetitions as described below"

Line 94 "injury outside the workplace" is listed as a covariate. It is assumed that this is injury to any part of the body other than the back?

Line 95 "Whereas the.." should be deleted. Sentence should start with "Discomfort.."

Line 113 onwards. Is this determined by an interviewer timing and watching the worker?

Line 127 "multivariate" should be "multivariable"

Line 128 "Lameshow" should be "Lemeshow"

Line 131 "counter" should perhaps be "determine"

Line 135 "insignificant" should be "not significant at p<0.05"

There are issues with English and clarity throughout the methods and this needs to be checked.

Results:

Table 2 reports SD not variance

Line 149 should be "multivariable" instead of "multivariate"

Line 151 "insignificant" should be "not significant" This needs to be changed throughout the document.

Line 155 It is probably preferable to report only Hosmer and Lemeshow test for goodness of fit rather than Omnibus test. Why are both beta coefficient and odds ratios reported? OR would be sufficient.

Paragraph between line 149 and 169 is not clear and needs reworking.

Paragraph between line 170 and 193 is also not clear and needs reworking.

"Research question 1 and 2 could be deleted from the headings of these 2 sections.

Line 182 Where is table 7.

Line 194 Prevalence of pain section. This section may be better placed before the logistic regression as it is descriptive.

The graph and table are not required, the table is perhaps more informative.

Discussion: Line 206 true prevalence will not be determined if those without back pain were excluded from the study.

Line 209 the word demonstrated is probably preferable to proved

Line 217 "probably because they belonged to different provinces of Pakistan" Is there evidence to support this?

Line 220 Table numbering is incorrect

Line 223 "Also, it does not have a positive value of the logistic coefficient B which makes it insignificant for the pain in upper back." This is not the correct interpretation of these results.

Line 228 what is meant by "different levels of work-related musculoskeletal disorders?"

Line 230 "Young workers are used to working for 13 hours a day while older workers were unable to work more than 4 or 6 hours a day" Is there evidence for this?

Line 263 "Also, no worker with any history of non-work-related back pain were found. Thus, the workplace was found to be responsible for the back pain experienced by workers." It is likely that workers did not have time to do other things but also according to the exclusion criteria these workers were excluded from the study, thus this statement can not be made.

The discussion is generally not clear and needs to be reworked

Limitations section. This should be reworked. Sampling was potentially biased, data are self-report, sample of each type of task was small, exclusion criteria may have been an issue.

Conclusion needs to be reworked.

Classification tables do not need to be reported.

Table is not required for goodness of fit test, this could be reported in the text or as a footnote to the main table. Inclusion of goodness of fit test more relevant rather than Omnibus test.

Table numbering is problematic.

Odds ratios should only be reported for logistic regression, constant should not be reported in the table, neither should B, Wald statistic or SE. There is incorrect interpretation of the statistics, all covariates for model 1 are significant p<0.05 not just those with a value of 0.000. E.g. a value of 0.014 is also significant at p<0.05. This is also supported by the ORs none of which cross 1. Reporting of the fitted model is not required unless was going to be used to predict pain presence.

Table heading should reflect what the model is assessing ie factors associated with lower back pain.

Tables B1-3 same comments as above.